# Characterization of a New Biofunctional, Exolytic Alginate Lyase from *Tamlana* sp. s12 with High Catalytic Activity and Cold-Adapted Features

**DOI:** 10.3390/md19040191

**Published:** 2021-03-28

**Authors:** Rui Yin, Yan-Jun Yi, Zhuo Chen, Bao-Xun Wang, Xue-Han Li, Yan-Xia Zhou

**Affiliations:** Marine College, Shandong University, Weihai 264209, China; yinrui@mail.sdu.edu.cn (R.Y.); yiyanjun@mail.sdu.edu.cn (Y.-J.Y.); chenzhuo-sdu@mail.sdu.edu.cn (Z.C.); wangbaoxun@mail.sdu.edu.cn (B.-X.W.); lixuehan@mail.sdu.edu.cn (X.-H.L.)

**Keywords:** alginate lyase, cold-adapted lyase, exolytic alginate lyase, bifunctional alginate lyase

## Abstract

Alginate, a major acidic polysaccharide in brown algae, has attracted great attention as a promising carbon source for biorefinery systems. Alginate lyases, especially exo-type alginate lyase, play a critical role in the biorefinery process. Although a large number of alginate lyases have been characterized, few can efficiently degrade alginate comprised of mannuronate (M) and guluronate (G) at low temperatures by means of an exolytic mode. In this study, the gene of a new exo-alginate lyase—Alys1—with high activity (1350 U/mg) was cloned from a marine strain, *Tamlana* sp. s12. When sodium alginate was used as a substrate, the recombinant enzyme showed optimal activity at 35 °C and pH 7.0–8.0. Noticeably, recombinant Alys1 was unstable at temperatures above 30 °C and had a low melting temperature of 56.0 °C. SDS and EDTA significantly inhibit its activity. These data indicate that Alys1 is a cold-adapted enzyme. Moreover, the enzyme can depolymerize alginates polyM and polyG, and produce a monosaccharide as the minimal alginate oligosaccharide. Primary substrate preference tests and identification of the final oligosaccharide products demonstrated that Alys1 is a bifunctional alginate lyase and prefers M to G. These properties make Alys1 a valuable candidate in both basic research and industrial applications.

## 1. Introduction

Alginate is the most abundant polysaccharide of brown algae [1]. It is a line unbranched hetero-polysaccharide consisting of 1,4-linked β-d-mannuronic acid (M) and α-L-guluronic acid (G), arranged in varying blocks of polyM, polyG, and the heteropolymer polyMG [2]. In recent years, alginate has received widespread attention as a promising carbon source for the production of biofuels and versatile biochemicals [1,2,3,4,5,6]. In the biorefinery process, alginate is firstly monomerized by alginate lyases, especially exo-type lyases, into an unsaturated uronate, which can be non-enzymatically converted to 4-deoxy-l-erythro-5-hexoseulose uronate (DEH) [4,5,6]. Then, with the presence of a series of reductases and kinases, DEH is converted to 2-keto-3-deoxy-6-phosphogluconate (KDPG), assimilated into the Entner–Dourdoroff (ED) pathway of industrial microorganisms, and is finally utilized to form bioethanol and other chemicals in biorefinery systems [4,5,6,7,8]. Thus, the degradation efficiency of alginate lyases is a prerequisite for the production of bioethanol and chemical compounds from brown algae. In addition, the depolymerization of alginates can balance the excess, reducing equivalents produced by the fermentation of mannitol [1,3], another rich monosaccharide of brown alga, so as to realize the full potential fermentation of brown algae to produce bio-based chemicals. As a result, the use of alginate lyases is critical for the saccharification and utilization of alginate and the effective use of brown algae. Alginate lyases catalyze the degradation of alginate by β-eliminating the glycoside 1, 4-o-glycoside bonds between C-4 and C-5 at the non-reducing end, resulting in the production of oligouronic acids or uronic acid monomers. Based on the catalytic mode, alginate lyases are divided into the endolytic and exolytic type. The exo-alginate lyases (EC 4.2.2.26) usually depolymerize unsaturated oligosaccharides produced by the action of endo-alginate lyases on alginate. Some exo-alginate lyases have been reported to directly monomerize alginate to a monosaccharide [1,9]. In terms of substrate specificities, alginate lyases are classified into three types—the G-specific type (guluronate lyases, EC 4.2.2.11), M-specific type (polyM lyase, EC 4.2.2.3) and a bifunctional type that exhibit activities towards both polyG and polyM [9,10,11]. Although a number of alginate lyases have been characterized and classified into twelve polysaccharide lyase (PL) families (PL5, 6, 7, 14, 15, 17, 18, 32, 34, 36, 39 and 41) in the Carbohydrate-Active Enzyme (CAZy) database, the identified exo-alginate lyases are still fewer in number, and most are classified into PL15, including six oligoalginate lyases. To date, the largest family of alginate lyases, PL7, includes just two exo-alginate lyase out of 38 alginate lyases. Alginate lyase AlyA5 of *Zobellia galactanivorans* DsijT [12] and alginate lyase Alg7K of *Saccharophagus degradans* [3,13] possess exolytic activity to polyM and polyG. Furthermore, exo-alginate lyase AlyA5 has been structured 3D model in detail and has been proven to act by means of a calcium-dependent mechanism, suggesting an exquisite adaptation to their natural origins (from bacteria or from brown sea-weeds) [12]. Alginate lyases with special characteristics are crucial to the conversion of brown algae biomass and present advantages in the industrial processes. In recent years, several alginate lyases have been discovered and reported, including the cold-adapted alginate lyases [12,14,15], thermo-tolerant alginate lyases [16], and high-alkaline alginate lyases [17]. Interestingly, due to high activity in the biocatalytic processes at low temperatures, the cold-adapted enzyme, a part of marine carbon cycling, saves energy and production costs, and also reduces the risk of microbial contaminations and product denaturation [15,18]. Therefore, cold-adapted alginate lyase has been considered to have great potential applications in industrial biorefineries. Unfortunately, most research focuses on endo-alginate lyases. Considering the lack of exo-alginate lyases with excellent characteristics, it is of great urgency to obtain cold-adapted biofunctional exo-alginate lyases.

In this study, the gene of a new alginate lyase—Alys1—was successfully cloned from *Tamlana* sp. s12 by means of genome mining. Then, it was heterologously expressed in *Escherichia coli* Rosetta (DE3). The recombinant Alys1 directly degraded the alginate to yield monosaccharides, disaccharides, trisaccharides and some products lower than monomers as main products. This study also revealed that polyM-preferred Alys1 possesses cold-adapted, biofunctional characteristics. These special features suggest that Alys1 may play essential roles in saccharification processes of alginate and marine carbon cycling.

## 2. Results

### 2.1. Sequence Analysis of Alys1

The marine bacterium *Tamlana* sp. s12 was isolated from the gut of edible holothurian *Apostichopus japonicus* at Weihai, China. It grew rapidly in the alginate sole-carbon medium and efficiently degraded the alginate, with a high alginate lyase activity. The open reading frame of gene *alys1*, amplified from the genome of *Tamlana* sp. s12, is 1017 bp in length and encodes an alginate lyase containing 355 amino acid residues (Genbank number OBQ55419), including a signal peptide of 23 residues (Met^1^ to Cys^23^). The theoretical isoelectric point (pI) and theoretical molecular weight (Mw) of the deduced Alys1 protein were 5.21 and 38.04 kDa, respectively. According to the CAZy databases, Alys1 is a new putative alginate lyase with a single-domain Met^52^-Ser^347^ belonging to the PL7, the largest family of alginate lyases. Furthermore, among the characterized alginate lyases, Alys1 shared the highest identity (71.8%) with an exo-alginate lyase, AlyA5 from *Z. galactanivorans* DsiJT (Genbank number AF208293). As was shown in the phylogenetic analysis (Figure 1), Alys1 formed a strong clade with several reported alginate lyases of subfamily 5, indicating that Alys1 is a novel member of the subfamily 5 of PL7. Moreover, protein sequence alignment showed that the Alys1 module contained one catalytic motif (QIH) and the critical residues Gln 150, His 152, and Tyr 302, which was conserved in structurally characterized PL7 alginate lyases (Figure 2). Alys1 also contained two highly conserved regions of RXEL and YFKXGXYXQ, which were characteristic of PL7. Thus, the predicted domain structure, the phylogenetic position, and the presence of essential sites all suggested that the deduced Alys1 is a novel putative alginate lyase of PL7.

### 2.2. Expression, Purification, and Characterization of Alys1

The full-length *alys1* gene without a signal peptide was amplified directly from the genomic DNA of *Tamlana* sp. strain s12. The 1-kbp PCR products were cloned into the pET-28a (+) vector downstream of a T7 promoter. A His6 tag was successfully added to the C terminus of the protein product (recombinant Alys1 (rAlys1)) in the expression vector (pET28-*alys*1). SDS-PAGE analyses indicated that *Rosetta agami* (DE3) cells harboring the recombinant plasmids pET28-*alys*1 could produce soluble proteins in the supernatant fraction (Figure 3a) with the correct apparent molecular masses. As was shown in Figure 3b, the sharp and distinct clearance zones occurred around the supernatant and sediment of the induced cell lysate of *Rosetta agami* (DE3) containing pET28-*alys*1, whereas the absence of clearance zones appeared around the induced cell lysate of *Rosetta agami* (DE3) containing pET-28a (+) and the uninduced cell lysate of *Rosetta agami* (DE3) containing pET28-*alys*1, with the exception of a small zone around the supernatant fraction of the latter due to the background expression. These results further proved that the rAlys1 is a soluble alginate lyase. A single band of purified rAlys1 was observed in the SDS-PAGE gel (Figure 3a). To investigate the state of the rAlys1 in solution, the purified rAlys1 was analyzed by the fast protein liquid chromatography (FPLC). Two peaks were obtained and indicated by the numbers 1 and 2 in the gel chromatography (Appendix A). Eluent fraction of the first peak lacked not only the enzyme activity of alginate lyase but also a band in SDS-PAGE profile (Appendix A), which indicated that it wasn’t an alginate lyase or a protein. Meanwhile, eluent fraction of the second peak possessed alginate lyase activity and exhibited a single band corresponding to a molecular weight of about 45 kDa in the SDS-PAGE gel (Appendix A), which was consistent with the purified rAlys1(Figure 3a). Thus, the second peak was considered to correspond to rAlys1, of which similar molecular weight calculated from gel filtration was about 44.6 kDa. Therefore, rAlys1 existed monomers in water without soluble aggregates. Subsequently, the purified rAlys1 was able to be further used for biochemical characterization.

As shown in the temperature profile (Figure 4a), rAlys1 exhibited the highest activity at 35 °C, above 50% of the maximum activity at 10 °C, and no detectable activity at 60 °C. Compared with other enzymes, rAlys1 had a wider temperature range for the activity. However, the activity of rAlys1 significantly reduced when the temperature exceeded 35 °C. We further investigated the thermostability of rAlys1. As shown in Figure 4b, the activity of rAlys1 was relatively unchanged after 1 h of incubation at 10 °C, rapidly inactivated at temperatures above 30 °C, and approximately 70% of its activity was lost after incubation at 50 °C for 60 min. Furthermore, rAlys1 remained more than 90% of its maximum activity at 20 °C, similarly with other cold-adapted enzymes from PL7 under the same conditions, such as AlyS02 from *Flavobacterium* sp. S02 and Alyw201 from *Vibrio* sp. W2 [19,20]. Moreover, the apparent Tm of rAlys1 determined by differential scanning calorimetry (DSC) was 56.0 °C (Figure 4c), which was consistent with the thermal stability of the rAlys1. DSC is mainly used to determine first-order transitions such as the denaturation of proteins [21]. Since the detector of DSC3 device was a power compensator, which was able to keep the temperature of rAlys1 sample close to that of the reference, an inert material without phase change as the temperature increases, by compensating the heating power on both sides in time, a compensation signal was received in the DSC curve. In other words, the endothermic peaks obtained for protein unfolding processes appeared as a downward peak in DSC curve [22,23]. The endothermic peak was formed during the melting process and the denaturation of the rAlys1 (Figure 4c). This process comprises the reversible or irreversible loss of the original ordered structure into a disordered arrangement of the polypeptide chains. Using the same substrate, cold-adapted enzymes usually have lower optimum temperature, higher activity at low temperatures, and lower thermostability compared to their mesophilic homologs [24,25,26,27,28]. These results indicated that the thermostability of rAlys1 is quite low, further supporting that rAlys1 is a cold-adapted enzyme.

Although alginate lyase rAlys1 had no detectable activity in 50 mM Tris-HCl, it showed activity in phosphate buffers. Alginate lyase rAlys1 exhibited the highest activity between pH 7.0–8.0 in 20 mM Na_2_HPO_4_–NaH_2_PO_4_ buffer and 20 mM Na_2_HPO_4_–KH_2_PO_4_ buffer (Figure 4d). Meanwhile, rAlys1 remained 97.1% of the highest activity at pH 7.0 and had a favorable stability between pH 7.0–8.0 in 20 mM Na_2_HPO_4_–NaH_2_PO_4_ buffer (Figure 4e). Its different activities in different phosphate buffers at the same pH suggested that buffer ions affected its activity. Further analysis showed that the rAlys1 activities were significantly inhibited by 1 mM NH_4_Cl, FeCl_3_, FeSO_4_, MnCl_2_, CaCl_2_, SDS, and EDTA, whereas its activities were slightly improved by 1 mM KCl, MgCl_2_, MgSO_4_, Glycerol and NaCl (Figure 4f). Notably, the enzyme activity of rAlys1 was only weakly affected by changing NaCl concentrations from 0.01 to 0.3 M (data not shown). The rAlys1 activity was increased with Na^+^, Mg^2+^, and K^+^, like alginate lyases from the PL7 family, which are essential for the enzyme activation for most alginate lyases [19].

In addition, the role of glycerol for rAlys1 was further investigated. As shown in Figure 4g, glycerol activated rAlys1 with concentration of 10–50%. With 20% glycerol, the activity of rAlys1 reached the highest. To evaluate the thermal stability in glycerol of rAlys1, the enzyme was incubated in 20% glycerol at different temperature. As shown in Figure 4h, the activities of rAlys1 were significantly improved by glycerol at 10, 35 and 60 °C, approximately 156–210% of relative activity compared to that without glycerol. These results indicated that glycerol assist in the stability of the rAlys1 during preservation at lower or higher temperature. Using sodium alginate as a substrate, alginate lyase rAlys1 had enzyme activity of 1350.51 ± 20.13 U/mg, *Km* values of 0.20 ± 0.01 mM, *Kcat* values of 4.43 ± 0.027 s^−1^, and *Kcat*/*Km* of 220.54 ± 5.49 s^−1^mM. The *Km* and *Kcat*/*Km* indicated relatively high substrate-binding affinity and catalysis efficiency of rAlys1.

### 2.3. Analysis Substrate Specificity and Final Products

To determine the substrate specificity, 1.0% alginate, polyG, and polyM were used as the substrates to study the activity of the enzyme. The relative activities of rAlys1 toward alginate, polyG, and polyM were 48.1 ± 2.2%, 71.0 ± 2.2% and 100.0 ± 6.2%, respectively, indicating that rAlys1 is a polyM-preferred bifunctional alginate lyase (Figure 5a). It has been reported that the substrate specificities of PL7 alginate lyases are related with the QIH protein sequences in the conserved regions. The polyG-specific and polyMG alginate lyases contain QIH in the conserved region [17,29,30]. However, recent studies have revealed that some alginate lyases containing the QIH sequence showed preferences for polyM [16,31,32,33]. As a polyM-preferred bifunctional alginate lyase, rAlys1 was in accordance with the latter. To investigate the action modes of rAlys1, the degradation products of alginate, polyG, and polyM after 0.5 h and 12 h were analyzed by means of thin-layer chromatography (TLC). As shown in Figure 5b, rAlys1 degraded substrates into mono- di-, tri-, tetra-oligosaccharides, and some products lower than monomers. The monomer products indicated that rAlys1 acts on alginate, polyG, and polyM exolytically. Based on the largest TLC spot size of the minimum products, rAlys1 had high activity toward polyM. The results were consistent with the enzymatic assays. However, the degradation products of substrates after 12 h were only detected monosaccharides and some products lower than monomers (Figure 5c). Furthermore, the distribution of final depolymerization products of alginate was monitored by liquid chromatography-mass spectrometry (LC-MS), which were only detected monosaccharides (Figure 5d). This result was consistent with the TLC analysis (Figure 5c). All the results illustrated that rAlys1 depolymerized alginate polymers in an exolytic manner. This indicated that rAlys1 could be a potential tool for the preparation of lower molecular weight products, which have wide pharmaceutical applications. In addition, it remains to be further determined whether the minimal product is DEH, which is easily utilized to form bioethanol and other chemicals in biorefinery systems.

### 2.4. Molecular Modeling of Alys1

The three-dimensional structure of Alys1 without a signal peptide was constructed using SWISS-MODEL [34] on the basis of homologs of known structure (AlyA5, PDB ID: 4BE3) with sequence identity of 76%. Alys1 is predicted to have the same jelly roll-fold and additional loops as AlyA5 [12]. The key residues for catalytic activity were predicted to be residues Q127, H129, and Y279 (Figure 6). Mutation of these residues were purified (Appendix A) and had no enzyme activity. 

## 3. Discussion

Alys1 is the third exolytic alginate lyase with special characteristics in the PL7 family. On the one hand, the results of sequence analysis showed that Alys1 had 71.8% (the highest) identity and conservative catalytic amino acids compared to AlyA5, a PL7 characterized alginate lyase from *Zobellia galactanivorans* DsiJT [12]. Based on functional characterization, both were exotype enzymes with broad substrate specificity, but their substrate preference is different. AlyA5 was reported to prefer polyG, whereas Alys1 preferred polyM. Hence, the conserved amino acids QIH in the catalytic cavity have little relation to substrate recognition of polyM or polyG. On the contrary, the loop around the catalytic cavity or the non-conservative amino acids may play a role in substrate recognition [12,35,36]. On the other hand, it was found that the enzyme activity of Alys1 at a low temperature of 10 °C reached 56% of the highest enzyme activity, and that Alys1 has low Tm and poor thermal stability, high physiological coefficient *Kcat*/*Km*, sensitivity to SDS and EDTA. These results indicate that Alys1 has some characteristics of a cold-adapted enzyme. The cold adaptation mechanism and modes of a wide range of substrates should be further clarified. 

## 4. Materials and Methods 

### 4.1. Strains, Plasmids and Chemicals

*Tamlana* sp. s12, preserved in our laboratory, was as used as a resource of genomic DNA for cloning the *alys1* gene. The genomic DNA of the s12 strain was sequenced and deposited into the GenBank database under the accession number LDKC01000006. *Escherichia coli* DH5α and the vector pUC19 (Sangon Biotech, Shanghai, China) were used for gene cloning. *E. coli* Rossate (DE3) (Solarbio, Beijing, China) and the vector pET-28a (+) (Invitrogen, America) were used for protein expression. *E. coli* strains all were grown in Luria–Bertani (LB) medium. Sodium alginate (M/G ratio: 1.66) was purchased from Qingdao Bright Moon Seaweed Group Co., Ltd. (Qingdao, China). Standard alginate monosaccharide, disaccharide, trisaccharide, tetrasccharide, polyG (M/G ratio 1.8/98.2, purity: 99%) and polyM (M/G ratio 97.3/2.7, purity: 99%) were purchased from Qingdao BZ Oligo Biotech Co., Ltd. (Qingdao, China). Other chemicals and reagents used in this study were of analytical grade. Unless otherwise stated, other reagents were of analytical or higher grade and were commercially available.

### 4.2. Cloning, Expression and Purification of Alys1

The alginte lyase gene *alys1* was amplified using the primers *alys1*-F (5′-TACTCAGGATTCTCATTAACTAGTTGCGTTAA-3′) and *alys1*-R (5′-TACTCAAAGCTTTTACTCCATATCGGGAGGTG-3′), containing *Bam*H I and *Hind* III restriction sites (underlined), respectively. After the PCR fragment was digested by *Bam*H I and *Hind* III, the purified fragment was cloned into pUC19 for sequencing, then subcloned into the pET-28a (+) expression vector and finally transformed into *E. coli* Rossate (DE3). The recombinant cells harboring pET-28a (+)-*alys*1 were grown in Luria–Bertani medium supplemented with kanamycin (50 μg/mL) at 37 °C and 180 rpm until the optical density at 600 nm reached 0.6, then induced by isopropyl-1-thio-β-d-galactoside (IPTG) at a final concentration of 1 mM at 20 °C and 140 rpm for 22 h. Then, cells were harvested by centrifugation at 4 °C and 8000 rpm for 10 min, resuspended in 50 mM phosphate buffer (pH 7.4) and broken by ultrasonication. After centrifugation at 12,000 rpm at 4 °C for 10 min twice, the supernatant was loaded onto a nickel-nitrilotriacetic acid (Ni-IDA) affinity column (Sangon Biotech, Shanghai, China). The purified fractions were collected, desalted using economical biotech membrane (Sangon Biotech, Shanghai, China) and further analyzed using 10% SDS-PAGE system (Bio-Rad, Hercules, CA, USA). The protein marker was purchased from Takara (Beijing, China). The purified rAlys1 was subjected to AKTA FPLC with a Superdex peptide 200 10/300 gel filtration column (GE Health, Marlborough, MA, USA) eluting in 20 mM Na_2_HPO_4_–NaH_2_PO_4_ (pH 8.0) at a flow rate of 0.5 mL/min. The reaction was monitored at 280 nm by the ultraviolet detection system. The peak-based elutions were concentrated using a Millipore centrifugal filter 10 K device (Millipore, Billerica, MA, USA). Gel Filtration Calibration kit Low Molecular Weight (GE Health, Marlborough, MA, USA) was used to create standardization plots by graphing the log_10_ molecular weight against *Ve*/*Vo* (elution volume/void volume) and predicting a protein molecular weight. The protein concentration was determined using the method of Bradford [37] and using bovine serum albumin (BSA) as the standard.

To rapidly and sensitively identify the activity of the alginate lyase, the plate assay of Gram’s iodine was improved [38]. The 0.9% agarose gel containing 1.0% alginate was flooded with a four-fold diluted solution of Gram’s iodine after 3 h incubation of alginate lyase at 35 °C. A clearance zone around the alginate lyase was distinct within 1–2 min after flooding the agarose gel.

### 4.3. Sequence Analysis of Alys1

For functional annotation of the predicted proteins, similarity searches of amino acid sequences were performed using the BLAST algorithm on the National Center for Biotechnology Information server (NCBI) (http://www.ncbi.nlm.nih.gov, accessed on 2 August 2020). Molecular weights of the putative proteins were estimated using the peptide mass tool on the ExPASy server of the Swiss Institute of Bioinformatics (http://swissmodel.expasy.org/, accessed on 2 August 2020). The theoretical pI of Alys1 were calculated with the compute pI Tool (https://web.expasy.org/compute_pi/, accessed on 2 August 2020). The signal peptide cleavage site of Alys1 was predicted through the SignalP 5.0 server (http://www.cbs.dtu.dk/services/SignalP/, accessed on 2 August 2020). Protein modules and domains were identified in the products using the Simple Modular Architecture Research Tool (https://en.wikipedia.org/wiki/Simple, accessed on 2 August 2020 Modular Architecture Research Tool), the Pfam database (http://pfam.xfam.org, accessed on 2 August 2020), the CAZy database (http://www.cazy.org, accessed on 2 August 2020), and the NCBI Conserved Domain Database (CDD) [39]. Multiple sequence alignments were carried out using DNAMAN version 9.0 (Lynnon biosoft, San Ramon, CA, USA). Phylogenetic analyses were performed using MEGA version 7.0 [40].

### 4.4. Enzyme Activity Assay

The activity of the alginate lyase was determined by 3,5-dinitrosalicylic acid (DNS) colorimetry [41]. One unit (U) of enzyme activity was defined as the amount of enzyme required to release 1 μmol of reducing sugar per min. Unless otherwise noted, the activity was measured at 35 °C for 30 min in a mixture of 80 µL buffer (20 mM Na_2_HPO_4_–NaH_2_PO_4_, pH 8.0), 100 µL alginate substrate (10 mg/mL), and 20 µL enzyme extract in triplicate. Then, 300 µL of 3,5-dinitrosalicylate (DNS) solution was added to the solution. After incubation, the mixture was boiled for 5 min to terminate the reaction. Finally, the absorbance was measured at 540 nm.

### 4.5. Characterization of rAlys1

Unless otherwise noted, sodium alginate was the substrate used in the enzyme activity assay for rAlys1 characterization. To determine the effect of temperature on rAlys1, enzyme reactions were carried out at different temperatures (10–60 °C). To evaluate the thermal stability of rAlys1, the enzyme was incubated for 1 h at 10–60 °C. Then, the residual activities of the enzyme were tested. The activity of the enzyme stored at 10 °C was used to represent 100% enzyme activity. The Tm of rAlys1 was characterized by differential scanning calorimetry. The purified rAlys1 (with 10 mM freeze-dried powder stabilizer) and 10 mM freeze-dried powder stabilizer were freeze-dried in 10 mM phosphate buffer (pH 7.0) to DSC on a differential scanning calorimeter (DSC3, METTLER TOLEDO, Zurich, Switzerland). Aluminum was used as a standard substance. The flow rate of shielding gas (N_2_) was 50 mL/min. Data were collected from 0.0 to 130.0 °C in 1.0 °C intervals.

The effect of different pH values on rAlys1 was determined by calculating the residual activities of rAlys1 after 20 h of incubation at 4 °C in 20 mM Na_2_HPO_4_–Citric acid (pH 3.0–6.0), Na_2_HPO_4_–NaH_2_PO_4_ (pH 6.0–7.5), or Na_2_HPO_4_–KH_2_PO_4_ (pH 7.5–9.0) buffers. The initial activities in different pH buffers represented 100% enzyme activity. To determine the effects of metal ions, glycerol, SDS and EDTA on the activity of rAlys1, the highest enzyme activity was considered as 100% enzyme activity. rAlys1 was subjected to an activity assay after 12 h of incubation at 4 °C in the presence of 1.0 mM of different metal ions, glycerol, SDS and EDTA. To determine the effect of glycerol on rAlys1, enzyme reactions were carried out at different concentration (10–50%). The activity was measured at 35 °C for 30 min in a mixture of 100 µL mixture of glycerol and buffer (20 mM Na_2_HPO_4_–NaH_2_PO_4_, pH 8.0), 80 µL alginate substrate (10 mg/mL), and 20 µL enzyme extract in triplicate. Then, the activities of the enzyme were tested. To evaluate the thermal stability of glycerol on rAlys1, the enzyme was incubated for 1 h at 10, 35, 60 °C in 20% glycerol. Then, the residual activities of the enzyme were tested. The activity of the enzyme stored at 10 °C was used to represent 100% enzyme activity.

### 4.6. Substrate Specificity, Degradation Products and Kinetic Parameters of rAlys1

In this study, 1% (*w*/*v*) substrate solutions (20 mM Na_2_HPO_4_–NaH_2_PO_4_, pH 8.0) with three kinds of polymers (sodium alginate, polyG and polyM) were used to measure the enzyme activities by the DNS method described above to assess the preferred substrate of rAlys1.

To determine the oligosaccharide compositions of the final digests, 30 µL diluted enzyme (5 µM) was added to 170 µL substrate solution containing 0.2% (*w*/*v*) substrate, 20 mM Na_2_HPO_4_ -NaH_2_PO_4_ buffer, and 200 mM NaCl (pH 7.0). After incubation at 35 °C for 0.5 and 12 h, the reaction buffer terminated by boiling for 5 min, and analyzed by TLC with the solvent system (1-butanol/ acetic acid/water 2:1:1). The TLC plate was sprayed using sulfuric acid/ethanol reagent (1:4, *v*/*v*), and then heated at 85 °C for 5 min. The mixture of monosaccharide, disaccharide, trisaccharide and tetrasccharide was applied as a marker, and the total uronic acid concentration was 0.2% (*w*/*v*). To further determine the composition of the end products, the degraded products were desalted and detected by LC-MS instrument (6230B HPLC/TOF, Agilent, USA) after incubation at 35 °C for 12 h. The oligosaccharides were detected in a positive-ion mode using the following settings: ion source voltage, 3.5 kV; capillary temperature, 325 °C; sheath gas, 8 L/min; scanning the mass range, 50–1500 *m*/*z*.

The kinetic parameters of the purified enzyme toward sodium alginate and polyM were determined by measuring the enzyme activity with substrates at different concentrations (0.1–8.0 mg/mL), as sodium alginate is a polymer consisting of random combinations of mannuronic acid and guluronic acid residues. Since they both have the same molecular weight (MW), substrate molarity was calculated using the MW of 176 g/mol for each monomer of uronic acid in the polymer. The concentrations of the product were determined by monitoring the increase in absorbance at 235 nm using the extinction coefficient of 6150 M^−1^cm^−1^. Velocity (V) at the tested substrate concentration was calculated as follows: V (mol/s) = (milliAU/min × min/60 s × AU/1000 milliAU × 1 cm)/(6150 M^−1^cm^−1^) × (2 × 10^−4^ L). The Km and Vmax values were calculated by hyperbolic regression analysis, as described previously [42]. Additionally, the turnover number (Kcat) of the enzyme was calculated by the ration of Vmax versus enzyme concentration ([E]). The kinetic parameters of rAlys1 toward alginate were determined by measuring the initial velocities of enzyme activity under various sodium alginate concentrations and were calculated on the basis of the nonlinear regression fitting of the Michaelis–Menten equation using Prism 6.0 (GraphPad Software, Inc., La Jolla, CA, USA).

### 4.7. Molecular Modeling and Site-Directed Mutagenesis of Alys1

The three-dimensional structure of Alys1 was constructed using SWISS-MODEL on the basis of homologs of known structure (AlyA5 from Z. galactanivorans DsijT with PDB ID: 4BE3) with sequence identity of 76%. PyMOL 2.2.0 software (DeLano Scientific LLC, San Carlos, CA, USA) was used for the analysis of the modeled structure and loops. Using the vector pUC19 as a temple, the site-directed mutagenesis of Q127, H129 and Y279 to rAlys1 was performed with the Site-directed Mutagenesis Kit (Sangon Biotech, Shanghai, China) by following the manufacturer’s protocol. The mutant primers were designed to amplify the DNA sequence, and the mutagenesis sites were underlined (Appendix A).

## 5. Conclusions

In this study, we reported a new alginate lyase derived from the marine strain *Tamlana* sp. s12. Alys1 was characterized as a novel PL7 exo-alginate lyase with cold-adapted features. Alys1 has poor Tm and thermal stability and possess a range of structural features, which was consistent with the cold-adapted alginate lyase from PL7 family. Although Alys1 contained the QIH sequence in the conserved regions, which was thought to be polyG-specific, Alys1 turned out to be polyM-preferred, with a broad range of substrate specificities. The characteristics of a cold-adapted exolytic alginate lyase with a broad range of substrate specificities have great significance for the use of alginates to produce biofuels and chemical compounds in biorefinery applications. Further work will be focused on other cold-adapted characters on cell level and the contribution of particular amino acids to elucidate cold-adapted and substrate recognition mechanism of Alys1. 

## Figures and Tables

**Figure 1 marinedrugs-19-00191-f001:**
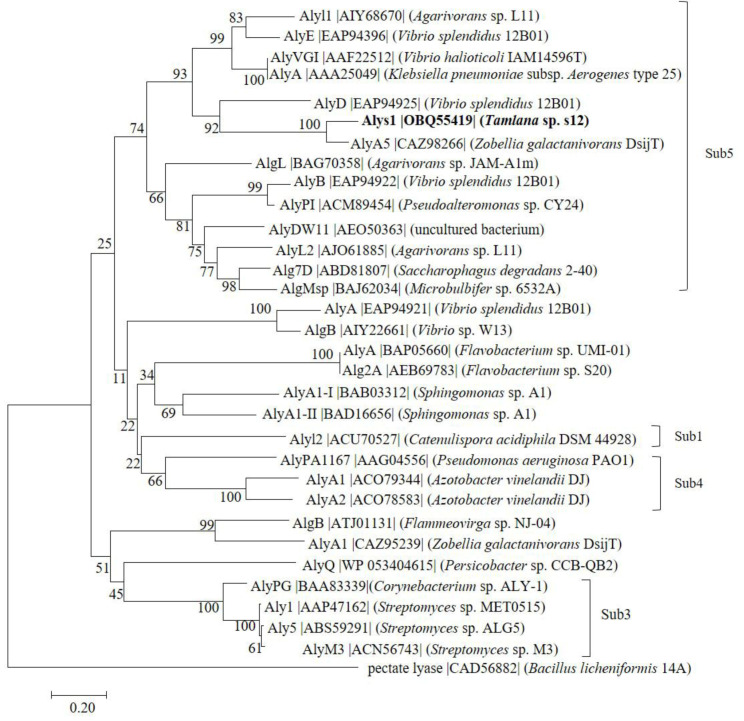
Phylogenetic analysis of Alys1 with other reported alginate lyases. The reliability of the phylogenetic reconstructions was determined by boot-strapping values (1000 replicates). Branch-related numbers are bootstrap values (confidence limits) representing the substitution frequency of each amino acid residue. A pectate lyase (CAD56882) from *Bacillus licheniformis* 14A was used as a control. The species names are indicated along with the accession number of the corresponding alginate lyase sequence. Bootstrap values of 1000 trials are presented in the branching points. Bar, 0.20 substitutions per nucleotide position.

**Figure 2 marinedrugs-19-00191-f002:**
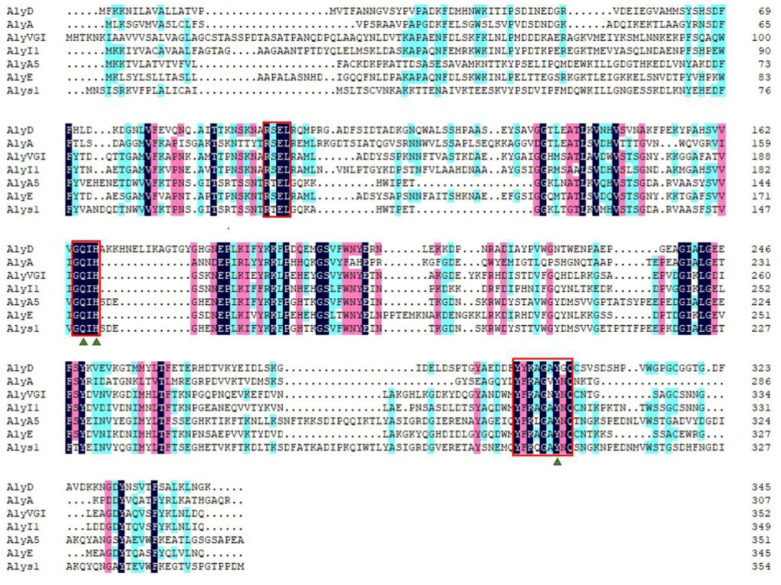
Multiple sequence alignments of alginate lyase Alys1 and related alginate lyases of the PL7 family. Alys1 (OBQ55419) from *Tamlana* sp. s12 in this study, AlyD (EAP94925) from *Vibrio splendidus* 12B01, AlyA (AAA25049) from *Klebsiella pneumoniae* subsp. *Aerogenes* type 25, AlyVGI (AAF22512) from *Vibrio halioticoli* IAM14596T, Alyl1 (AIY68670) from *Agarivorans* sp. L11, AlyA5 (CAZ98266) from *Zobellia galactanivorans* DsiJT, AlyE (EAP94396) from *Vibrio splendidus* 12B01. The conserved amino acid regions are highlighted with red boxes. The potential residues involved in the catalytic activity in the PL7 family are indicated with green triangles.

**Figure 3 marinedrugs-19-00191-f003:**
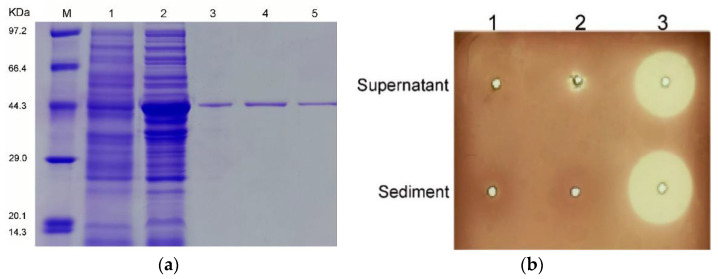
Expression and purification of rAlys1. (**a**) SDS-PAGE analysis of purified protein rAlys1. M, protein marker; column 1, induced cell lysate of *Rosetta agami* (DE3) containing plasmid pET28a (+); column 2, supernatant fraction of induced *Rosetta agami* (DE3) containing pET28-*alys1* before purification; columns 3–5, eluent fraction after purification. (**b**) Plate assay for alginate lyase active of the recombinant protein rAlys1 using Gram’s iodine. Column 1, induced cell lysate of *Rosetta agami* (DE3) containing pET28a (+); column 2, uninduced cell lysate of *Rosetta agami* (DE3) containing plasmid pET28-*alys1*; column 3, induced cell lysate of *Rosetta agami* (DE3) containing plasmid pET28-*alys1*.

**Figure 4 marinedrugs-19-00191-f004:**
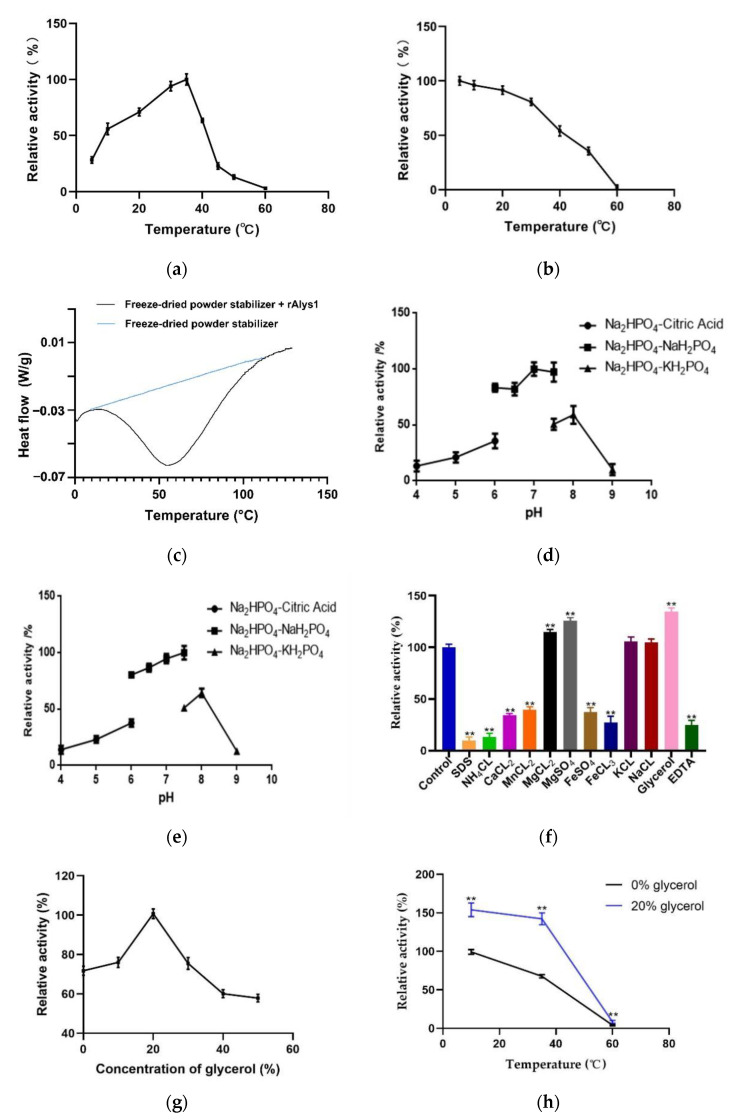
The biochemical characteristics of alginate lyase rAlys1. (**a**) The optimal temperature of alginate lyase rAlys1; (**b**) the thermal stability of alginate lyase rAlys1; (**c**) the melting temperature (Tm) of rAlys1. Black line, the purified rAlys1 with 10 mM freeze-dried powder stabilizer; blue line, 10 mM freeze-dried powder stabilizer; (**d**) the optimal pH of alginate lyase rAlys1 in 20 mM Na_2_HPO_4_–Citric acid buffer (filled circle), 20 mM Na_2_HPO_4_–NaH_2_PO_4_ buffer (filled square), or 20 mM Na_2_HPO_4_–KH_2_PO_4_ buffer (filled triangle); (**e**) the pH stability of the alginate lyase rAlys1 in 20 mM Na_2_HPO_4_–Citric acid buffer (filled circle), 20 mM Na_2_HPO_4_–NaH_2_PO_4_ buffer (filled square), or 20 mM Na_2_HPO_4_–KH_2_PO_4_ buffer (filled triangle); (**f**) the effect of different metal ions, EDTA, Glycerol and SDS on Alys1, ** denote the significant differences (*p* < 0.01) from the control; (**g**) the optimal glycerol concentration of rAlys1; (**h**) the thermal stability of alginate lyase rAlys1 in glycerol, ** denote the significant differences (*p* < 0.01) from the 0% glycerol.

**Figure 5 marinedrugs-19-00191-f005:**
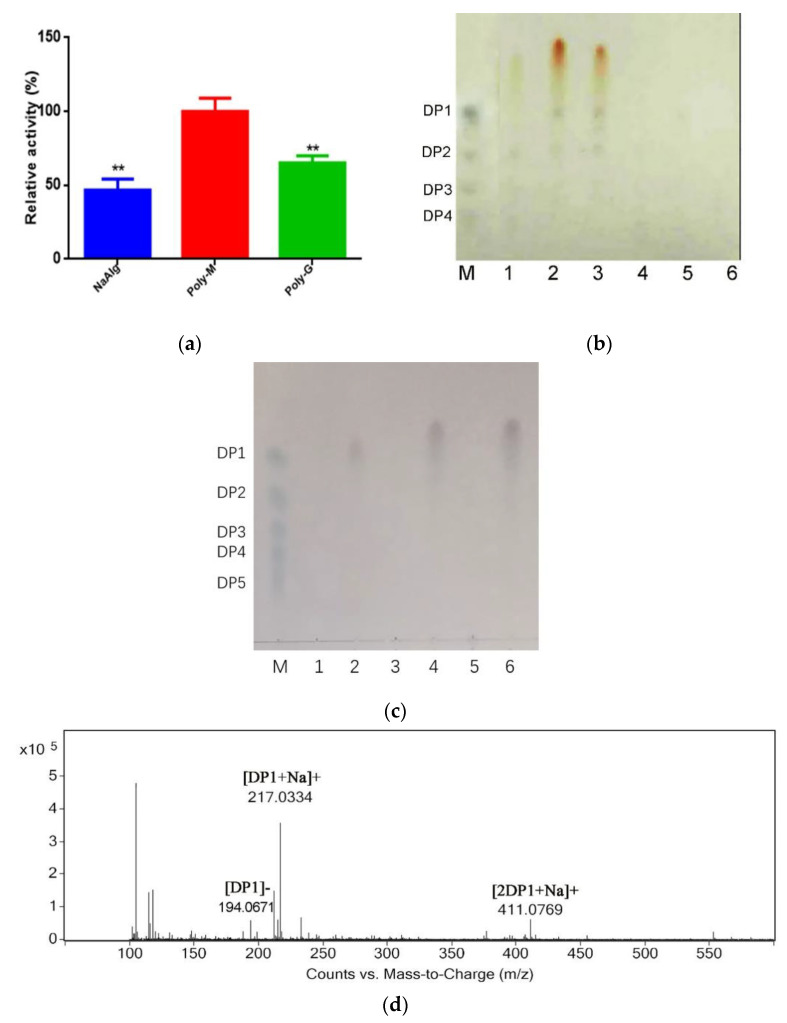
The substrate specificity and final products of alginate lyase rAlys1. (**a**) Relative enzyme activities of rAlys1 a toward alginate, polyM, and polyG, ** denote the significant differences (*p* < 0.01) from the poly-M. (**b**) TLC analysis of the degradation products of alginate lyase rAlys1 toward alginate, polyM, and polyG for 0.5 h. Column M, reference standards; column 1, products of alginate sodium; column 2, products of Poly-M; column 3, products of Poly-G; columns 4–6, culture medium containing 0.2% alginate sodium, Poly-M and Poly-G. (**c**) TLC analysis of the degradation products of alginate lyase rAlys1 toward alginate, polyG, and polyM for 12 h. Column M, reference standards; columns 1, culture medium containing 0.2% alginate sodium; column 2, products of alginate sodium; columns 3, culture medium containing 0.2% Poly-G; column 4, products of Poly-G; column 5, culture medium containing 0.2% Poly-M; column 6, products of Poly-M. (**d**) LC-MS analysis of the degradation products of alginate lyase rAlys1 toward alginate for 12 h.

**Figure 6 marinedrugs-19-00191-f006:**
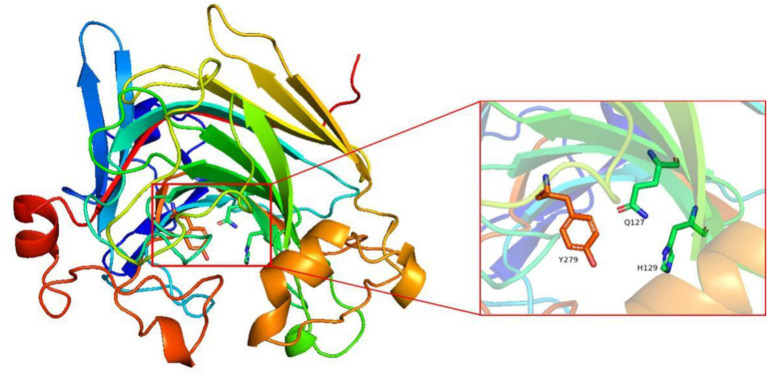
Three-dimensional structures of Alys1. The key residues for substrate specificities in Alys1 are shown as sticks model.

## Data Availability

Data is contained within the article or Appendix A.

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
