# Peer review of "Characterization of a New Biofunctional, Exolytic Alginate Lyase from *Tamlana* sp. s12 with High Catalytic Activity and Cold-Adapted Features"

_marinedrugs, 2021, doi:10.3390/md19040191_

Round 1

Reviewer 1 Report

Alginase degrades alginate, a major acidic polysaccharide that is a potential carbon source for biorefinery systems. The authors describe the characterization of a recombinant exolytic alginate lyase (Alys1) from Tamlana sp. S12. The recombinant enzyme displayed optimal activity at 35˚ C, but retains more than 50% of its activity at 10˚C. This enzyme is capable of depolymerizing alginates poly-mannuronate (M) and poly-guluronate (G) at low temperatures, but prefer polyM. The properties of this alginase makes it an ideal candidate for industry.

Specific Comments:

(1) The authors carefully examined the effect of a number of salts on activity. Did the authors investigate the role of glycerol to increase activity at lower or higher temperatures? This may also assist in the stability of the enzyme during purification.

(2) As Mg2+ is essential for the activation of most alginate lyases, can the authors speculate why it may inhibit the activity of their enzyme?

(3) How reduced was the activity of the enzyme variants in which the residues Q127, H129, and Y279 were altered to alanine?

Reviewer 2 Report

The work entitled “Characterization of a new cold-adapted, biofunctional and exolytic alginate lyase from Tamlana sp. s12 with high catalytic activity” by Yin et al., reports on the cloning of the ORF coding for a new putative alginate lyase from the bacterium Tamlana sp. s12, the production and purification of the recombinant enzyme and finally its biochemical characterization.

I think that this is a nice piece of work, technically solid. The manuscript is very well written and the scientific message is very clear. In my opinion this work is quite interesting for the potential readers of this journal. Nevertheless, before its publication authors should address several major points that I saw with some concern after careful reading. There are some minor points that I indicate in the pdf also.

Major points:

  • One major concern is related to the classification of rAlys1 as a cold-adapted enzyme. I start out by saying that this should not be a relevant aspect of the work since the experimental results are what they are independently of how we define them, either as coming from a psychrophilic or a mesophilic enzyme. I see some bias of the authors in trying to define rAlys1 as cold-adapted and this is not necessary. In this regard, authors provide some structural characteristics of rAlys1 that would support the hypothesis that it is a cold-adapted enzyme. However, the presence of these characteristics does not unambiguously define an enzyme as cold-adapted since the contribution of particular amino acids to the global stability of the protein should be analyzed in detail for each one and not in a coarse-grained mode. Currently, there are many enzymes from psychrophiles that do not fit those rules and in fact, highly thermostable enzymes have been described from psychrophilic organisms. An additional aspect of this topic is related to the fact that cold-adaptation should be necessarily associated to the cellular level. For instance, the obtained results shown in figure 4b may indicate that rAlys1 is a thermolabile enzyme, which is considered typical of cold-adaptation. In these experiments rAlys1 is preincubated for 1 h at the indicated temperature and then its activity tested (I suppose at 35 ºC). However, the question arises as to which is the half-live of the enzyme in the cell? In any case, as I indicated before, I consider the classification of rAlys1 as cold-adapted should not be a relevant aspect of this work. Authors should therefore relax their claim that it is cold-adapted.
  • DSC results (figure 4c). I am afraid I am not familiar with the DSC3 device and therefore I might be wrong but according to the experimental section thermal unfolding of rAlys1 has been studied with a freeze-dried sample and not with a sample of protein in solution. This fact makes comparison of these results with those obtained with protein in solution (activity assays) highly unreliable. In fact, if I am right, the registered peak showed in figure 4c is exothermic, which is at odds with the endothermic peaks obtained for protein unfolding processes in solution. Of course, protein thermal unfolding entails exothermic contributions mainly due to the disruption of hydrophobic interactions but the overall process is always endothermic. There are also exothermic contributions, usually after the main endothermic peak, due to the aggregation and precipitation of the denatured protein. Authors should provide an explanation of the obtained result for rAlys1 since this is important for the determination of the apparent melting temperature of rAly1.
  • Purification of rAlys1. According to the Materials and Methods section, rAlys1 has been purified by IMAC, considering that it possesses a C-terminal His-tag. Desalted protein sample is used for the analyses. Although SDS-PAGE (figure 3a) indicate that rAlys1 is highly purified, authors do not provide information about the state of the protein in solution. In particular, for a rigorous analysis of the biochemical characteristics of rAlys1, authors should provide information about the stability of the enzyme in solution: does rAlys1 samples present aggregates? This is quite important since presence of soluble aggregates clearly influence the activity results. Gel-filtration is mandatory for this type of analyses.

Round 2

Reviewer 2 Report

Authors have properly addressed the points I raised* and therefore in my opinion this new version of the manuscript has improved importantly. The results from gel-filtration analysis are correct and add relevant information to the work. However, just before publication, authors should tackle these points:

  • *Only one point remains unclear to me, namely, DSC results. Now, authors included (page 6; lines 206-207): “The first endothermic peak was formed during the melting process and the denaturation of the rAlys1”. However, as I indicated previously: “… if I am right, the registered peak showed in figure 4c is exothermic, which is at odds with the endothermic peaks obtained for protein unfolding processes in solution”. So, the author´s claim remains unclear to me. Nevertheless, I insist that I am not familiar with that device that uses freeze-dried samples.
  • Although authors have relaxed the claim that Alys1 is a cold-adapted enzyme all along the manuscript, the bias towards it is cold-adapted remains in the title, which is disproportionate with respect to the text. Therefore, I suggest to change the title to: “Characterization of a biofunctional, exolytic alginate lyase from Tamlana s12 with high catalytic activity and cold-adapted features”, which is more coherent.
